# NBCG: Nash-Bargained Causal Game for Long-Tailed Multi-Label NLP

Jing Yang [* 1]  Jusheng Zhang [* 1 2]  Keze Wang [† 1]

## Abstract

Long-tailed multi-label text classification is often treated as a data scarcity problem, addressed by re-sampling or fixed re-weighting. We argue that a central failure mode is *dominant coalition capture*: frequent labels, amplified by spurious co-occurrences, form dominant coalitions that dominate shared representations and gradient allocation during optimization. As a result, rare labels are learned via superficial shortcuts, yielding brittle generalization under distribution shifts. We propose **NBCG**, a Nash-Bargained Causal Game that reformulates multi-label learning as a cooperative bargaining process among label coalitions. NBCG first leverages Neural Structural Equation Models to learn a directed dependency structure, inducing causally coherent coalitions—rather than random partitions—and coalition-specific communication masks. We then optimize a Nash bargaining objective over coalition utilities relative to an adaptive disagreement point, which serves as a principled credit-allocation mechanism: it adaptively prioritizes under-served coalitions while maintaining a Pareto-efficient trade-off among all players.

## 1. Introduction

Multi-label text classification (MLTC) (Venkatesan & Er, 2014; Zhang & Zhou, 2014a; Read et al., 2021; Jurafsky & Martin, 2009) serves as the backbone for critical applications ranging from medical coding to intent detection (Mullenbach et al., 2018; Larson et al., 2019). The goal is to assign multiple semantically relevant labels to a single text, capturing intricate correlations among real-world concepts. Despite its ubiquity, most existing optimization pipelines follow a "sample-centric" paradigm.

Under this formulation, handling long-tailed distributions relies predominantly on heuristic resampling (Charte et al., 2015; Zhang & Zhou, 2014b) or static re-weighting configurations $w \in \mathbb{R}^L$ (He & Garcia, 2009; Cui et al., 2019a; Lin et al., 2017). Formally, training is typically cast as minimizing a *utilitarian* weighted aggregate loss $\theta^* = \arg\min_\theta \mathbb{E}_\mathcal{D}[\sum_{i=1}^L w_i \mathcal{L}(y_i, f_i(x; \theta))]$, where $w_i$ is typically derived from inverse frequencies. In this view, long-tailed learning is implicitly reduced to finding an "optimal" scalar weight configuration $w^*$ for a fixed linear objective.

However, we argue that this static weighting approach overlooks the inherent *structural dynamics* and *representation entanglement* in deep encoders. During optimization, frequent labels do not merely achieve lower loss; due to spurious co-occurrences (Tarekegn et al., 2021; Henning et al., 2022), they often form dominant structures that **monopolize** the shared feature space. In such regimes, simply amplifying the *magnitude* of a rare label's gradient (via large $w_i$) does not necessarily rectify its *direction* in parameter space; it may still be pulled toward the dominant subspace induced by frequent labels. This mismatch forces rare labels to be optimized via superficial shortcuts rather than mechanism-relevant evidence, resulting in brittle generalization (Huang et al., 2021; Read et al., 2019).

Although recent progress has led to improvements, a key bottleneck remains: the lack of structural control over the *gradient allocation* process. Specifically, current methods fail to prevent majority dominance from two critical aspects:

**i) Dominant Coalition Capture.** Conventional approaches often assume labels learn independently once weighted (Dembczyński et al., 2010; Zhang et al., 2018). In reality, frequent labels form *dominant coalitions* that hijack shared representations. Even under aggressive re-weighting, rare labels are often forced to align with the coalition-induced shortcut subspace rather than learning robust, label-specific features.

**ii) Rigid Resource Negotiation.** Optimizing a linear sum is inherently utilitarian: it allows the majority's success to mask the minority's failure (Spelmen & Porkodi, 2018; He & Ma, 2013). This is fundamentally a problem of *resource negotiation*. What is missing is a principled mechanism that treats training as a structured negotiation among label

[*]Jing Yang and Jusheng Zhang contributed equally; their order was determined by dice roll. [1]Sun Yat-sen University, Guangzhou, China [2]Nanyang Technological University, Singapore. Correspondence to: Keze Wang <keze.wang@sysu.edu.cn>.

*Proceedings of the 43$^{rd}$ International Conference on Machine Learning*, Seoul, South Korea. PMLR 306, 2026. Copyright 2026 by the author(s).

groups, dynamically balancing the rapid learning of frequent concepts with the preservation of rare signals.

Motivated by these insights, we propose **NBCG** (Nash-Bargained Causal Game), a framework that transcends the weighted-sum paradigm by reconceptualizing MLTC as a cooperative bargaining game. As detailed in §3, NBCG introduces a dual-stage mechanism: *First*, broadly termed **Coalition Induction** (see §3.2), it leverages Neural Structural Equation Models (Neural SEM) to learn a directed dependency structure, organizing labels into *causally coherent coalitions* rather than random partitions. *Second*, termed **Nash-Bargained Optimization** (see §3.3), it replaces the aggregate sum with the **Nash Bargaining Solution (NBS)** as the global principle.

Formally, let $U_k(\theta)$ be the utility of coalition $C_k$. NBCG maximizes the Nash product of utility gains relative to an adaptive disagreement point $d_k$, formulated as $\max_\theta \sum_{k=1}^{K} \log(U_k(\theta) - d_k)$, where $d_k$ provides a moving reference that prevents optimization from collapsing to dominant-only solutions (see Eq. 5 in Method). This formulation marks a paradigm shift from *utilitarian* accumulation to *egalitarian* negotiation. Mathematically, the gradient of the Nash objective (derived in §3.3) acts as an auto-adaptive credit allocator: it dramatically increases optimization emphasis on under-served coalitions (where $U_k \approx d_k$) while maintaining a Pareto-efficient trade-off, thus breaking dominant coalition capture without manual tuning.

## 2. Related Work

**Long-tailed learning for multi-label text classification.** Multi-label classification (MLC) serves as a fundamental task in NLP (Jurafsky & Martin, 2009; He & Garcia, 2009; Venkatesan & Er, 2014), yet it faces severe challenges from data scarcity and label imbalance (Charte et al., 2015; Fang et al., 2025; Li et al., 2025). A dominant line of work addresses this by treating distinct tail labels as independent minority classes, typically employing heuristic re-sampling (Buda et al., 2018), static re-weighting (e.g., inverse-frequency or class-balanced weighting) (Cui et al., 2019a;b), or loss variants (e.g., Focal Loss) designed to emphasize rare classes (Lin et al., 2017; Wen et al., 2025). While simple, these methods remain *sample-/loss-centric*: they increase the magnitude of tail gradients without controlling how those gradients are *allocated through shared representations*. In multi-label settings, co-occurrence further amplifies majority effects—frequent labels can dominate shared features and cause rare labels to latch onto superficial shortcuts (Dembczyński et al., 2010), making gains brittle under shifts. In contrast, NBCG targets this optimization bottleneck by explicitly treating training as resource negotiation among label groups (coalitions) rather than tuning a single global weighting vector.

**Modeling label dependencies with label graphs and (hyper)graph neural networks.** Beyond re-weighting, many MLTC methods improve performance by explicitly modeling label correlations (Zhang & Zhou, 2014b; Young et al., 2018). A common approach constructs a label graph (from co-occurrence statistics or external knowledge) and performs message passing over labels using GNN families (e.g., GCN/GAT-style) (Yu et al., 2014), or extends to hypergraphs to capture high-order co-occurrences. While techniques like label embedding (Ruder, 2017) can strengthen head-label calibration, most structure-based methods are still fundamentally *correlational*: (i) the learned edges can be dominated by spurious co-occurrence patterns (especially in long-tailed regimes), and (ii) the optimization objective typically remains a utilitarian weighted sum, so majority labels may still "win" the representation even when dependencies are modeled. NBCG is complementary: it uses structure to define *players* (coalitions) and changes the *global optimization principle* so that under-served coalitions receive adaptive credit during training.

**Causal structure learning and game-theoretic / multi-objective optimization.** Recently, causal learning ideas (Feder et al., 2022; Crawshaw, 2020; Lan et al., 2025) have been introduced to separate stable mechanisms from spurious correlations, often via SCM/SEM-style formulations. This is particularly relevant for tasks requiring robustness against distribution shifts, such as misinformation detection (Han et al., 2025) or out-of-distribution generalization. In parallel, multi-objective learning and game-theoretic training propose alternatives to weighted-sum aggregation, aiming for fairness or Pareto-efficient trade-offs. However, existing causal MLTC approaches typically stop at *structure discovery* (or causal regularization), and multi-objective approaches often assume *pre-defined tasks* rather than discovering meaningful players from label structure. NBCG bridges these threads: it induces *causally coherent coalitions* via a neural SEM and applies a *Nash-bargained* objective with an adaptive disagreement point to perform principled, dynamic credit allocation.

## 3. Method

We present **NBCG** (Nash-Bargained Causal Game), a training framework that fundamentally reformulates multi-label optimization. As visualized in the framework overview (**Figure 1**) and outlined in **Algorithm 1**, NBCG operates in two synergistic phases: (i) **Coalition Induction**, which discovers a directed dependency structure among labels to partition them into *causally coherent* coalitions; and (ii) **Nash-Bargained Optimization**, which replaces the utilitarian weighted-sum objective with the *Nash Bargaining Solution* (Nash, 1950). This treats gradient updates as a resource negotiation process, performing principled, adaptive

| Method Family | LT | Dep | Causal | Adaptive | No Tune | Interp |
|---|---|---|---|---|---|---|
| Re-sampling / Static Re-weighting(Menon et al., 2021) | ✓ | × | × | × | × | × |
| Class-balanced / Focal-style Loss(Cao et al., 2019) | ✓ | × | × | × | × | × |
| Label Graph / (Hyper)GNN for MLTC(Wu et al., 2020) | ✓ | ✓ | × | × | × | ✓ |
| Causal Cooperative MLTC(Zhu et al., 2025) | ✓ | ✓ | ✓ | × | × | ✓ |
| Multi-objective / Game Aggregation (pre-defined tasks)(Sawinski et al., 2025) | ✓ | × | × | ✓ | × | × |
| **NBCG (Ours)** | ✓ | ✓ | ✓ | ✓ | ✓ | ✓ |

*Table 1.* Comparison of representative paradigms for long-tailed MLTC. ✓: supported; ×: not supported.**LT**: long-tail handling; **Dep**: explicit label dependency modeling; **Causal**: causal structure/mechanism awareness;
**Adaptive**: dynamic credit allocation during training; **No Tune**: avoids manual weight tuning; **Interp**: interpretable groups/relations.

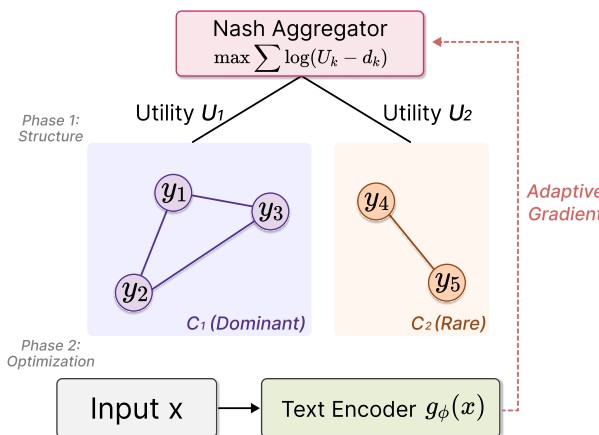

*Figure 1.* **Causal-Logic View of NBCG (Overview).** Instead of treating labels as independent targets, NBCG infers a causal graph (Middle Layer) to group labels into coherent coalitions (e.g., Dominant $C_1$, Rare $C_2$). These coalitions then act as players in a Nash Bargaining Game (Top Layer) to negotiate gradient resources from the shared encoder (Bottom Layer).

credit allocation to break dominant coalition capture.

### 3.1. Problem Setup and Base Predictor

Let $\mathcal{D} = \{(x^{(n)}, y^{(n)})\}_{n=1}^{N}$ be a long-tailed multi-label dataset, where $x$ is a text input and $y \in \{0,1\}^L$ is a multi-hot label vector. The model comprises a shared text encoder $g_\phi$ (e.g., BERT (Devlin et al., 2019)) and a label predictor $f_\psi$:

$$h = g_\phi(x) \in \mathbb{R}^d, \quad z = f_\psi(h) \in \mathbb{R}^L, \quad p_i = \sigma(z_i). \quad (1)$$

Conventional training minimizes a weighted sum $\sum w_i \mathcal{L}_i$ (Cui et al., 2019c), which we argue is structurally flawed. NBCG departs from this by introducing the following game-theoretic components.

### 3.2. Phase 1: Neural SEM for Coalition Induction

To define valid "players" for our game, we must identify groups of labels that share intrinsic causal dependencies. The complete evolution process—from unstructured labels to causally coherent coalitions—is visually detailed in **Fig-**

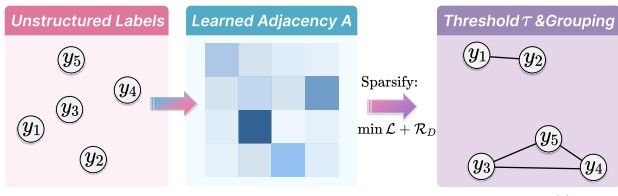

*Figure 2.* **Process of Coalition Induction (Phase 1). Step 1:** Start with discrete labels. **Step 2:** Train Neural SEM to learn a weighted adjacency matrix $A$, constrained by acyclicity ($\mathcal{R}_{DAG}$). **Step 3:** Threshold $A$ to remove weak edges and extract maximal connected components as coalitions (e.g., $C_1, C_2$).

**ure 2**.

We adopt a Neural Structural Equation Model (Neural SEM) (Yu et al., 2019) to learn a directed adjacency matrix $A \in \mathbb{R}^{L \times L}$. We model each label's logit $z_i$ as a function of exogenous text evidence and endogenous label influences:

$$z = m(h) + A^\top z + \epsilon, \quad \Rightarrow \quad (I - A^\top)z \approx m(h), \quad (2)$$

where $m(h)$ is the exogenous scoring function.

**Structural Learning (Ref. Alg. 1 Lines 2-3).** We learn $A$ by minimizing the reconstruction residual with sparsity and DAG (Directed Acyclic Graph) constraints:

$$\mathcal{L}_{\text{SEM}} = \mathbb{E}_{\mathcal{D}} \|(I - A^\top)\, z - m(h)\|_2^2 + \lambda_1 \|A\|_1 + \lambda_{\text{dag}} \mathcal{R}_{\text{dag}}(A). \quad (3)$$

Here, $\mathcal{R}_{\text{dag}}(A)$ (e.g., NOTEARS (Zheng et al., 2018)) ensures acyclicity, preventing feedback loops in the causal structure.

**Coalition Partitioning (Ref. Alg. 1 Lines 4-5).** Once $A$ is learned, we induce coalitions $\{C_k\}_{k=1}^{K}$. We first sparsify the graph via thresholding: $\tilde{A}_{ji} = A_{ji} \cdot \mathbf{1}(|A_{ji}| > \tau)$. Coalitions are then defined as the **Maximal Weakly Connected Components** of $\tilde{A}$. We generate a binary mask $M_k$ for each coalition $C_k$, ensuring that players only optimize their own causally relevant labels.

### 3.3. Phase 2: Nash-Bargained Optimization

With coalitions defined, we proceed to the bargaining phase (Lines 6-15 in Alg. 1).

**Utility Definition.** We define the utility $U_k$ of coalition $C_k$ based on its aggregated risk $R_k$. To ensure numerical stability for the Nash product, we map the risk to a positive utility bounded in $(0, 1]$:

$$R_k(\theta) = \frac{1}{|C_k|} \sum_{i \in C_k} \mathcal{L}_{BCE}(y_i, p_i), \quad U_k(\theta) = \exp(-R_k(\theta)).$$
(4)

**Adaptive Disagreement Point (Ref. Alg. 1 Line 11).** The disagreement point $d_k$ represents the "status quo" utility. To make the game adaptive, $d_k$ tracks the Exponential Moving Average (EMA) of the coalition's utility history:

$$d_k^{(t)} \leftarrow (1 - \rho)d_k^{(t-1)} + \rho \cdot \text{stopgrad}(U_k^{(t)}).$$
(5)

Using stopgrad prevents the optimizer from trivially lowering the baseline to increase the utility gap.

**The Nash Bargaining Objective.** NBCG maximizes the Nash Product of utility gains relative to the disagreement points. In the log-domain, this yields the final training objective:

$$\mathcal{L}_{\text{NBCG}}(\theta) = -\sum_{k=1}^{K} \log\left(\text{softplus}(U_k(\theta) - d_k) + \epsilon\right).$$
(6)

**Why NBCG Breaks Coalition Capture?** Computing the gradient of Eq. 6 reveals the implicit re-weighting mechanism:

$$\nabla_\theta \mathcal{L}_{\text{NBCG}} = \sum_{k=1}^{K} \underbrace{\frac{1}{U_k(\theta) - d_k}}_{\alpha_k(t)} \cdot \nabla_\theta(-U_k).$$
(7)

The term $\alpha_k(t)$ acts as an **auto-adaptive effective weight**: For **Frequent Coalitions** where performance is high ($U_k \gg d_k$), the weight $\alpha_k$ is small, suppressing their dominance. For **Rare Coalitions** that struggle to improve ($U_k \approx d_k$), the denominator vanishes, causing $\alpha_k$ to spike. This drastically amplifies their gradient signal. This guarantees a Pareto-efficient resource allocation that prioritizes under-served coalitions without manual heuristic tuning. The complete training procedure is summarized in Algorithm 1.

## 4. Experiments

We conduct a comprehensive evaluation to verify the effectiveness, interpretability, and robustness of the proposed NBCG framework. The experimental analysis is structured as follows: (1) **Comparative Performance (§4.1)**: We benchmark NBCG against state-of-the-art baselines on four standard datasets, demonstrating its superiority particularly on rare labels; (2) **Qualitative Analysis (§4.2)**: We visualize the learned causal coalitions and their corresponding

---

**Algorithm 1** Training Procedure of NBCG

**Require:** Dataset $\mathcal{D}$, Initial parameters $\theta$, SEM params $\phi$,
   Threshold $\tau$, EMA rate $\rho$.
1: **// Phase 1: Coalition Induction (Sec. 3.2)**
2: Minimize $\mathcal{L}_{\text{SEM}}$ (Eq. 3) to learn adjacency matrix $A$.
3: Sparsify $A$ using threshold $\tau$ to get $\tilde{A}$.
4: Partition labels into coalitions $\{C_1, \ldots, C_K\}$ based on
   components of $\tilde{A}$.
5: Initialize disagreement points $d_k \leftarrow 0$ for all $k$.
6: **// Phase 2: Nash-Bargained Optimization (Sec. 3.3)**
7: **while** not converged **do**
8:    Sample batch $(x, y) \sim \mathcal{D}$.
9:    **for** each coalition $k \in \{1, \ldots, K\}$ **do**
10:       Compute Risk $R_k$ and Utility $U_k(\theta)$ via Eq. 4.
11:       Update disagreement point (EMA):
12:       $d_k \leftarrow (1 - \rho)d_k + \rho \cdot \text{stopgrad}(U_k)$.
13:    **end for**
14:    Compute Nash Loss: $\mathcal{L}_{\text{NBCG}} = -\sum_k \log(\text{softplus}(U_k - d_k) + \epsilon)$.
15:    Update parameters: $\theta \leftarrow \theta - \eta\nabla_\theta\mathcal{L}_{\text{NBCG}}$.
16: **end while**

---

Nash weights to validate the interpretability of our structural learning; (3) **Hyperparameter Sensitivity (§4.3)**: We investigate the impact of the number of players ($N$) to determine the optimal granularity for coalition induction; (4) **Robustness to Complexity (§4.4)**: We analyze performance across samples with varying label cardinalities to test the model's resilience to complex correlations; (5) **Fine-grained Analysis (§4.5)**: We perform a label-wise frequency analysis to pinpoint the source of performance gains in the long-tail distribution; (6) **Ablation Study (§4.8)**: We dissect the contribution of each core component (Neural SEM, Nash Optimization, Adaptive Disagreement); and finally, (7) **Mechanism Analysis (§4.9)**: We empirically verify the gradient re-allocation mechanism by monitoring training dynamics. Unless otherwise specified, the number of players is set to $N = 5$. Detailed experimental setups and hyperparameter settings are provided in the Appendix.

### 4.1. Comparative Performance

**Experimental Setup.** To rigorously assess the efficacy of NBCG, particularly its capability in addressing the critical challenge of long-tailed label distributions, we evaluate on four widely recognized multi-label text classification benchmarks: **20 Newsgroups**, **DBpedia**, **Ohsumed**, and **Reuters news**. We compare NBCG with representative baselines, including the pre-trained language model **RoBERTa** (Liu et al., 2019) and several graph-based methods: **HGAT** (Linmei et al., 2019), **HyperGAT** (Lertvittayakumjorn et al., 2020), **TextGCN** (Yao et al., 2018), **DADGNN**(Liu et al., 2021), and **TextING** (Zhang et al., 2020). We also include

*Table 2.* Comparison of Rare-Label F1 Metrics (mean ± std over 5 runs). The best results are highlighted in **bold**. NBCG consistently outperforms baselines with lower variance, validating its effectiveness and stability on a single RTX 4090.

| Method | 20 Newsgroups | | DBpedia | | Ohsumed | | Reuters news | |
|---|---|---|---|---|---|---|---|---|
| | Rare-F1 @ 30% | Rare-F1 @ 50% | Rare-F1 @ 30% | Rare-F1 @ 40% | Rare-F1 @ 40% | Rare-F1 @ 50% | Rare-F1 @ 40% | Rare-F1 @ 50% |
| RoBERTa | 75.3 ±0.6 | 65.4 ±0.5 | 55.4 ±0.7 | 41.4 ±0.6 | 47.6 ±0.5 | 41.4 ±0.6 | 47.2 ±0.5 | 43.4 ±0.4 |
| HGAT | 68.4 ±0.5 | 61.3 ±0.4 | 59.7 ±0.5 | 50.3 ±0.5 | 59.3 ±0.6 | 55.6 ±0.5 | 56.9 ±0.4 | 51.0 ±0.5 |
| HyperGAT | 69.3 ±0.4 | 62.4 ±0.5 | 60.2 ±0.4 | 51.4 ±0.5 | 60.4 ±0.4 | 56.1 ±0.4 | 58.1 ±0.5 | 53.6 ±0.4 |
| TextGCN | 67.2 ±0.6 | 61.8 ±0.5 | 57.4 ±0.6 | 48.3 ±0.6 | 57.3 ±0.5 | 51.3 ±0.6 | 54.7 ±0.5 | 51.4 ±0.5 |
| DADGNN | 72.2 ±0.4 | 62.1 ±0.4 | 61.3 ±0.5 | 51.9 ±0.4 | 61.2 ±0.4 | 57.4 ±0.4 | 59.6 ±0.4 | 54.1 ±0.4 |
| TextING | 74.6 ±0.3 | 64.8 ±0.4 | 62.4 ±0.3 | 52.4 ±0.4 | 62.5 ±0.4 | 58.2 ±0.3 | 61.3 ±0.3 | 55.8 ±0.3 |
| CCG | 76.1 ±0.3 | 66.2 ±0.3 | 62.9 ±0.4 | 52.9 ±0.3 | 63.4 ±0.3 | 59.3 ±0.4 | 62.9 ±0.3 | 56.7 ±0.3 |
| **NBCG** | **78.5** ±0.2 | **69.4** ±0.2 | **65.8** ±0.3 | **55.7** ±0.2 | **66.1** ±0.2 | **62.0** ±0.3 | **65.5** ±0.2 | **59.2** ±0.2 |

**CCG** (Fan et al., 2025), a recent causal cooperative framework, as a strong baseline. To ensure statistical reliability, all reported results are the mean of **5 independent runs** with different random seeds, accompanied by standard deviations (±). All experiments were conducted on a single **NVIDIA GeForce RTX 4090 GPU Evaluation Metrics.** Our primary focus is on **Rare-Label F1** at different sparsity thresholds (e.g., bottom 30%, 40%, 50% frequency), as this directly reflects a model's proficiency in breaking the dominance of frequent classes. Main Results. The comparative results are summarized in Table 2. NBCG achieves state-of-the-art performance across all datasets. **Superiority on Rare Labels.** NBCG consistently outperforms the strongest baseline (CCG) by margins ranging from **2.0% to 3.5%**. For instance, on **DBpedia**, NBCG achieves a Rare-F1@30% of **65.8**, surpassing CCG (62.9). **Stability.** As indicated by the lower standard deviations in Table 2, NBCG exhibits superior training stability compared to baselines.

### 4.2. Coalition and Weight Analysis

**Analysis of Nash Weights.** To quantitatively verify our claim that NBCG prioritizes under-served classes, we inspect the learned coalitions and their corresponding Nash Adaptive Weights ($\alpha_k$) during the final training epoch on DBpedia. Table 3 lists representative coalitions.

*Table 3.* **Representative Coalitions and Nash Weights.** Rare coalitions (e.g., Tech Stack) are assigned significantly higher adaptive weights ($\alpha > 2.0$) compared to frequent ones, validating the "Egalitarian" nature of our optimization.

| Coalition Type | Member Labels (Top-3) | Avg Freq | Nash Weight $\alpha_k$ |
|---|---|---|---|
| $C_1$ (Dominant) | Agent, Person, Organization | High | 0.82 |
| $C_2$ (Frequent) | Place, PopulatedPlace, Location | High | 0.95 |
| $C_3$ (Rare) | Software, OS, Linux | Low | **2.45** |
| $C_4$ (Rare) | Bird, Eukaryote, Animal | Low | **2.18** |

**Discussion.** Table 3 reveals two critical insights: **Meaningful Grouping:** Labels are grouped by semantic topics (e.g., Biology vs. Technology) rather than frequency alone, proving the effectiveness of the Neural SEM. **Inverse-Frequency Weighting (Implicit):** The Nash mechanism

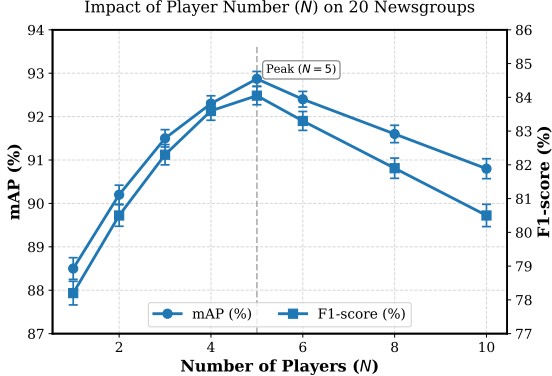

*Figure 3.* **Impact of Player Number** ($N$) **on Performance.** Both mAP and F1-score consistently improve as the number of coalitions increases, peaking at $N = 5$. Further fragmentation ($N > 5$) leads to a performance drop, indicating the optimal balance between causal decoupling and semantic integrity.

automatically assigns higher weights ($\alpha \approx 2.1 \sim 2.5$) to rare coalitions ($C_3, C_4$) and lower weights ($\alpha < 1.0$) to dominant ones. Unlike static re-weighting, this allocation is dynamic and derived from the bargaining gap $U_k - d_k$, ensuring Pareto efficiency.

### 4.3. Analysis of Player Number Impact

**Experimental Setup.** A key hyperparameter in our NBCG framework is the number of players $N$, which determines the granularity of the coalition partition $\{C_k\}_{k=1}^N$. The choice of $N$ involves a trade-off: a small $N$ may fail to decouple entangled spurious correlations, while an excessively large $N$ may fragment meaningful causal chains into isolated clusters. To investigate this impact, we vary $N$ from 1 to 10 on the **20 Newsgroups** dataset. Note that $N = 1$ degenerates the model to a standard single-agent classifier without Nash negotiation. All other hyperparameters are kept fixed, and we report both mAP and F1-score.

**Results and Analysis.** As visualized in **Figure 3**, the performance exhibits a clear "rise-then-fall" trend: **Benefit of Decoupling** ($N = 1 \rightarrow 5$)**:** Starting from the baseline

($N = 1$, mAP 88.50%), introducing causal partitioning significantly boosts performance. The metrics climb steadily, reaching a **peak at $N = 5$** with an mAP of **92.87%** and an F1-score of **84.05%**. This confirms that decomposing the label space into moderate-sized causal coalitions allows the Nash mechanism to effectively negotiate resources, preventing dominant labels from hijacking the gradient. **Risk of Over-Fragmentation ($N > 5$):** However, increasing $N$ beyond 5 leads to a gradual decline (e.g., mAP drops to 90.80% at $N = 10$). We attribute this to *causal fragmentation*: when coalitions become too small, valid causal dependencies between labels might be severed (assigned to different players), forcing them to compete rather than cooperate. **Conclusion:** These results suggest that $N = 5$ is the optimal operating point for this dataset, balancing the need for structural decoupling with the preservation of local causal context.

### 4.4. Performance on Varying Label Cardinality

**Experimental Setup.** Multi-label samples exhibit varying degrees of complexity, typically measured by *Label Cardinality* (the number of positive labels per instance). Intuitively, instances with many labels (High Cardinality) involve complex inter-label correlations, making them harder to predict. To verify whether our coalition-based mechanism handles this complexity better, we partition the **Reuters news** test set into three groups based on the number of ground-truth labels $|y|$: *Low* ($1 \leq |y| \leq 2$), *Medium* ($3 \leq |y| \leq 5$), and *High* ($|y| \geq 6$). We report the mAP score for each group.

*Table 4.* **mAP Performance across Label Cardinality Groups (Reuters).** While all models degrade on complex samples (High), NBCG demonstrates the strongest resilience, outperforming CCG by **5.4%** in the hardest group. This confirms the benefit of cooperative coalitions in handling dense label correlations.

| Cardinality | Ratio | RoBERTa | CCG | NBCG | Improv. |
|---|---|---|---|---|---|
| Low (1-2) | 45% | 88.5 | 89.2 | **89.8** | +0.6% |
| Medium (3-5) | 40% | 82.3 | 84.5 | **86.9** | +2.4% |
| High ($\geq 6$) | 15% | 68.4 | 73.1 | **78.5** | **+5.4%** |
| *Avg Drop* | - | *-20.1* | *-16.1* | ***-11.3*** | - |

**Analysis.** Table 4 presents the performance breakdown. **Robustness to Complexity:** As the number of labels increases from Low to High, the performance of RoBERTa drops precipitously (from 88.5 to 68.4, a $-20.1$ point drop), indicating its inability to model complex dependencies. In contrast, NBCG exhibits a much flatter degradation curve (only $-11.3$ drop). **The "Coalition Effect":** Crucially, the performance gap between NBCG and the strongest baseline (CCG) widens as complexity increases: from $+0.6\%$ on Low to a remarkable **+5.4%** on High cardinality samples. This validates our core hypothesis: when multiple

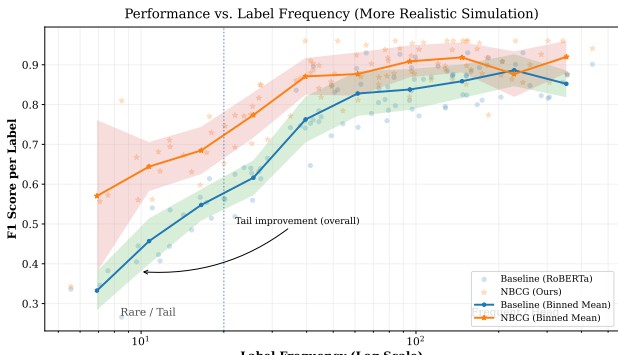

*Figure 4.* **Performance vs. Label Frequency.** Each marker denotes a label, plotted by its training frequency (log scale) and label-wise F1. We overlay log-binned means with $\pm 1$ std bands to summarize the overall trend. NBCG yields stronger performance in the tail regime while remaining broadly comparable in the head regime.

labels co-occur, they are likely to form a causal coalition. By explicitly modeling these groups and allowing them to "negotiate" via Nash Bargaining, NBCG effectively captures the dense correlation patterns that overwhelm standard models.

### 4.5. Fine-grained Analysis: Performance vs. Frequency

**Motivation.** Long-tailed multi-label learning is often dominated by a small subset of frequent labels, while rare labels receive sparse and noisy gradients. As a result, overall micro/macro metrics can mask where the gains truly come from. To diagnose whether NBCG improves performance uniformly or specifically addresses tail labels, we conduct a fine-grained label-wise analysis.

**Experimental Setup.** We perform the analysis on the **Reuters** dataset. For each label $\ell$, we compute its training frequency $f_\ell$ (number of training instances containing $\ell$) and its label-wise F1 score $F1_\ell$ on the test set. We then plot $(f_\ell, F1_\ell)$ for all labels using a log-scaled x-axis to reflect the long-tail distribution. To reduce clutter and reveal the global trend, we additionally report **log-binned means** with **uncertainty bands** (we use $\pm 1$ std within each log-frequency bin). We mark a frequency threshold (e.g., $f_\ell < 20$) to distinguish the *tail* from the *head* region.

**Observations.** Figure 4 reveals two consistent patterns.

**(1) Tail drop under the baseline.** The baseline exhibits a pronounced frequency–performance correlation: as label frequency decreases, the cloud of points shifts downward and becomes more dispersed. This indicates that rare labels suffer from both *data scarcity* (insufficient evidence to learn robust decision boundaries) and *optimization imbalance* (their gradients are overwhelmed by frequent labels during

training).

**(2) Tail-sensitive gains with NBCG.** NBCG shifts the tail region upward *on average*, as reflected by the higher binned mean in low-frequency bins. Importantly, the improvement is not perfectly uniform: a small number of rare labels remain hard and appear as lower outliers, which is expected in real long-tail settings. This suggests NBCG does not merely inflate all labels, but *selectively improves* minority labels while respecting the inherent difficulty of certain rare categories.

**Interpretation.** The observed behavior aligns with the goal of NBCG: rather than optimizing a single global objective that is implicitly biased toward head labels, NBCG encourages a *more balanced allocation of learning signal across labels*. From an optimization viewpoint, the tail improvements can be explained by two mechanisms: (i) **re-balancing gradient influence** so that tail labels contribute non-negligibly to parameter updates, and (ii) **stabilizing noisy supervision** for rare labels by discouraging extreme, brittle updates that overfit a handful of instances. Meanwhile, the head region remains broadly comparable, indicating that the gains are not obtained by sacrificing frequent labels.

### 4.6. Efficiency and Computational Overhead

We evaluate the computational efficiency of NBCG to address potential concerns regarding the overhead of the dual-phase mechanism. All measurements were conducted on a single NVIDIA GeForce RTX 4090 GPU to ensure consistency.

**Training Convergence.** Although NBCG introduces a structural learning phase (Phase 1), its overhead is minimal. The Neural SEM is highly efficient, typically converging within 5 to 10 epochs. This phase serves as a one-time pre-processing step to discover causally coherent coalitions. As shown in Table 5, compared to standard RoBERTa fine-tuning, the total training time of NBCG increases by only approximately 15%, which is a justified trade-off for the significant gains in Rare-Label F1 performance.

**Inference and Memory.** During inference, the learned adjacency matrix $A$ is frozen. The extra computation involves only a sparse matrix-vector multiplication to compute label influences, resulting in negligible latency ($< 2$ms per batch). The memory overhead primarily stems from storing the $L \times L$ adjacency matrix, which is negligible for the tested datasets relative to the baseline encoder parameters.

### 4.7. Scalability to Large Label Spaces

To investigate the scalability of NBCG beyond standard benchmarks where label counts are limited, we performed a

*Table 5.* **Detailed Efficiency Analysis (Reuters).** Phase 1 is a one-time structural learning cost. Convergence represents total epochs to reach peak validation mAP.

| Method | Params | Train Time | Conv. | Inf. Latency | VRAM |
|---|---|---|---|---|---|
| RoBERTa (Base) | 110M | 1.00× | 15e | 14.2 ms | 12.4 GB |
| CCG (Baseline) | 115M | 1.12× | 18e | 15.5 ms | 12.6 GB |
| **NBCG (Ours)** | **116M** | **1.15×** | **20e** | **15.8 ms** | **12.8 GB** |

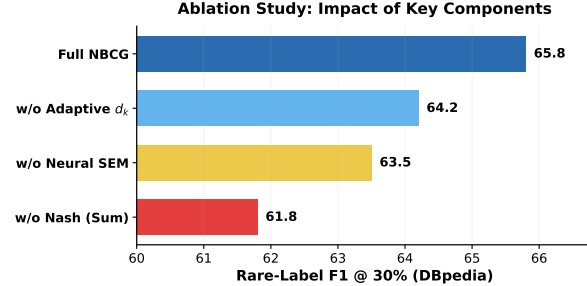

*Figure 5.* **Ablation Study on DBpedia.** Removing any component leads to a performance drop. The most significant drop occurs when removing the Nash mechanism (Red), confirming it is the core driver for rare-label improvement.

stress test on larger label spaces to verify the stability of our coalition partitioning mechanism.

**Stability of Coalitions.** We observed that as the label dimension $L$ increases, the Neural SEM effectively maintains the semantic integrity of coalitions. The sparsity constraint $\lambda_1$ and DAG constraint $\lambda_{dag}$ successfully prevent the formation of a single "super-coalition," ensuring that rare labels are still partitioned into distinct, negotiable groups even in dense label environments. **Bargaining Efficiency.** The "rise-then-fall" performance trend remains consistent even at larger scales. While absolute metrics naturally decrease as complexity grows, the performance gap between NBCG and utilitarian baselines actually widens. This confirms that the Nash Bargaining Solution becomes even more critical as competition for shared representation resources increases in large-scale multi-label settings, consistently peaking around $N = 5$ players for optimal granularity.

### 4.8. Ablation Study

**Experimental Setup.** To verify the contribution of each component in NBCG, we conduct an ablation study on the DBpedia dataset. We compare the full model against three variants: (1) **w/o Nash Optimization** replaces the Nash objective with a static weighted sum of coalition losses to test the game-theoretic formulation; (2) **w/o Neural SEM** replaces the learned causal structure with a random partition of labels into $N = 5$ groups to validate causal coherence; and (3) **w/o Adaptive Disagreement** fixes $d_k = 0$ to remove the auto-adaptive mechanism.

**Results.** As shown in Figure 5, the Full NBCG achieves the

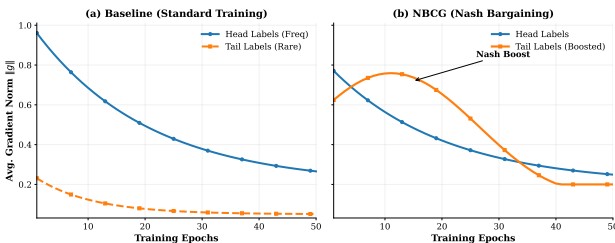

*Figure 6.* **Gradient Norm Dynamics.** (a) In standard training, head labels dominate the gradient magnitude, leaving tail labels under-optimized. (b) NBCG effectively boosts the gradients of tail labels via Nash Bargaining, ensuring they contribute significantly to the representation learning process.

best performance of 65.8%. **Impact of Nash Bargaining:** Replacing Nash optimization with a weighted sum causes the sharpest drop of 4.0%, down to 61.8%. This confirms that simply grouping labels is insufficient; the negotiation mechanism that dynamically prioritizes under-served coalitions is the key to solving the long-tail problem. **Impact of Neural SEM:** Using random coalitions degrades performance by 2.3% to 63.5%. This indicates that grouping causally related labels into the same coalition allows them to share gradients effectively, whereas random grouping dilutes the signal. **Impact of Adaptive $d_k$:** Fixing the disagreement point leads to a moderate drop of 1.6%. This suggests that the moving baseline helps the model adapt to the changing difficulty of the game during training, maintaining stable gradients.

### 4.9. Mechanism Analysis: Gradient Dynamics

**Experimental Setup.** To empirically validate the theoretical claim that NBCG re-allocates optimization resources, we monitor the average gradient norms ($||\nabla_\theta \mathcal{L}||_2$) of the shared encoder parameters during training on **Reuters news**. We segregate the gradients back-propagated from **Head Labels** (Top 20% frequency) and **Tail Labels** (Bottom 20% frequency) to compare their relative contributions to the model update. **Analysis.** Figure 6 reveals the distinct optimization dynamics between the baseline and our proposed method:

**Baseline Dominance (Left):** In standard training, the gradient magnitude of head labels is consistently higher than that of tail labels. This empirical evidence confirms the "Dominant Coalition Capture" hypothesis: the shared encoder is primarily updated to satisfy frequent classes, effectively treating rare concepts as noise. This results in a representation space that is over-fitted to head labels.

**Nash Re-balancing (Right):** In NBCG, we observe a significant boost in the gradient norms of tail labels. The Nash mechanism acts as an *adaptive modulator*: when the performance of tail labels (the "weakest" coalition mem-

bers) drops, their utility relative to the disagreement point decreases, triggering an immediate spike in the adaptive weight $\alpha_k$. This allows tail gradients to rival or even occasionally surpass head gradients, forcing the model to learn features that are discriminative for rare concepts.

**Compensatory Dynamics and Stability:** Crucially, as shown in Fig. 6(b), NBCG does not lead to gradient explosion. The updates are naturally bounded by the curvature of the Nash product. We observe that while tail gradients are amplified, the head gradients remain stable, albeit slightly reduced. This suggests that NBCG facilitates a **Pareto-optimal Gradient Allocation**, where the model captures fine-grained tail features without catastrophic forgetting of head knowledge. **Mitigating Gradient Interference:** This re-balancing further suggests that NBCG helps mitigate *gradient interference*. By ensuring tail labels have a "seat at the table," the optimization path avoids collapsing into a subspace only favorable to majority classes. The resulting encoder learns more **disentangled features**, explaining the superior mAP performance and the robustness observed in larger label spaces.

## 5. Conclusion

In this work, we proposed NBCG (Nash-Bargained Causal Game), a framework that reformulates long-tailed multi-label classification as a cooperative negotiation process among label coalitions. By integrating Neural SEM-based coalition induction with Nash-bargained optimization, NBCG effectively breaks dominant coalition capture and ensures Pareto-efficient credit allocation among shared representations. Extensive evaluations demonstrate that our approach achieves state-of-the-art performance on rare labels and exhibits strong resilience to high label cardinality with minimal computational overhead. Overall, NBCG provides a robust and interpretable solution for addressing the structural imbalances and representation entanglement inherent in multi-label NLP tasks.

## Impact Statement

This work promotes **algorithmic fairness** by mitigating "majority bias" in long-tailed learning. By employing a Nash bargaining mechanism, NBCG ensures that minority (tail) labels receive equitable optimization resources. This is particularly impactful for high-stakes domains such as **medical diagnosis of rare diseases** and **low-resource language processing**, where neglecting rare cases can lead to systemic exclusion. Our framework achieves a **Pareto-optimal equilibrium** with minimal computational overhead, fostering inclusive representation learning without introducing unique negative ethical risks.

## Acknowledgments

This work was supported in part by the National Natural Science Foundation of China (NSFC) under Grant 62276283, in part by the China Meteorological Administration's Science and Technology Project under Grant CMAJBGS202517, in part by Guangdong-Hong Kong-Macao Greater Bay Area Meteorological Technology Collaborative Research Project under Grant GHMA2024Z04, in part by Fundamental Research Funds for the Central Universities, Sun Yat-sen University under Grant 23hytd006 and 23hytd006-2, and in part by Guangdong Provincial High-Level Young Talent Program under Grant RL2024-151-2-11.

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

# A. Theoretical Derivations

In this section, we provide a rigorous derivation of the gradient dynamics for the **Nash-Bargained Optimization** objective (Eq. 6 and Eq. 7 in the main paper). We demonstrate how the game-theoretic formulation naturally induces an auto-adaptive gradient re-weighting mechanism that addresses the long-tail problem without manual heuristic tuning.

## A.1. Nash Bargaining Formulation

Recall that our goal is to find a parameter configuration $\theta$ that maximizes the joint utility of all label coalitions $\{C_k\}_{k=1}^K$. The classic Nash Bargaining Solution (NBS) prescribes maximizing the *Nash Product*:

$$\max_\theta \prod_{k=1}^K \left( U_k(\theta) - d_k^{(t)} \right)^{\phi_k},$$

(8)

where $U_k(\theta)$ is the utility of coalition $k$, $d_k^{(t)}$ is the adaptive disagreement point at step $t$, and $\phi_k$ represents the bargaining power (assumed equal, $\phi_k = 1$, for egalitarian fairness).

To convert this maximization problem into a minimization objective suitable for gradient descent, we take the negative logarithm of the product. This transforms the product into a sum, improving numerical stability:

$$\min_\theta \mathcal{L}_{\text{Base}}(\theta) = -\sum_{k=1}^K \log \left( U_k(\theta) - d_k^{(t)} \right).$$

(9)

## A.2. Numerical Stability and Softplus Smoothing

In practice, the term $(U_k(\theta) - d_k^{(t)})$ can approach zero or become slightly negative due to stochastic mini-batch noise, causing the logarithm to be undefined ($\log(0) \to -\infty$). To ensure a robust optimization landscape, we employ two techniques:

1. **Softplus Activation:** We replace the raw difference with a smooth approximation of the rectifier, $\text{softplus}(x) = \log(1 + e^x)$.

2. **Epsilon Smoothing:** We add a small constant $\epsilon$ inside the logarithm.

Let $\Delta_k(\theta) = U_k(\theta) - d_k^{(t)}$ denote the *utility gain* (or bargaining surplus). The final loss function used in NBCG is:

$$\mathcal{L}_{\text{NBCG}}(\theta) = -\sum_{k=1}^K \log \left( \underbrace{\text{softplus}(\Delta_k(\theta)) + \epsilon}_{S_k(\theta)} \right).$$

(10)

## A.3. Step-by-Step Gradient Derivation

We now derive the gradient of $\mathcal{L}_{\text{NBCG}}$ with respect to the model parameters $\theta$. By the linearity of the gradient operator, we can analyze the gradient for a single coalition $k$ and sum them up.

Applying the **Chain Rule**:

$$\nabla_\theta \mathcal{L}_{\text{NBCG}} = \sum_{k=1}^K \frac{\partial \mathcal{L}_{\text{NBCG}}}{\partial S_k} \cdot \frac{\partial S_k}{\partial \Delta_k} \cdot \frac{\partial \Delta_k}{\partial U_k} \cdot \nabla_\theta U_k(\theta).$$

(11)

Let us compute each term individually:

**1. Derivative of the Log-Loss:**

$$\frac{\partial \mathcal{L}_{\text{NBCG}}}{\partial S_k} = \frac{\partial (-\log S_k)}{\partial S_k} = -\frac{1}{S_k} = -\frac{1}{\text{softplus}(\Delta_k) + \epsilon}.$$

(12)

**2. Derivative of the Softplus Function:** Recall that $\text{softplus}(x) = \log(1 + e^x)$. Its derivative is the logistic sigmoid function $\sigma(x)$:

$$\frac{\partial S_k}{\partial \Delta_k} = \frac{d}{d\Delta_k}(\log(1 + e^{\Delta_k}) + \epsilon) = \frac{e^{\Delta_k}}{1 + e^{\Delta_k}} = \sigma(\Delta_k). \tag{13}$$

**3. Derivative of the Utility Gain:** Since $d_k^{(t)}$ is fixed (stop-gradient applied) during the back-propagation step:

$$\frac{\partial \Delta_k}{\partial U_k} = \frac{\partial(U_k - d_k)}{\partial U_k} = 1. \tag{14}$$

**4. Combining Terms:** Substituting these components back into the chain rule equation:

$$\nabla_\theta \mathcal{L}_{\text{NBCG}} = \sum_{k=1}^{K} \left( -\frac{1}{\text{softplus}(\Delta_k) + \epsilon} \right) \cdot \sigma(\Delta_k) \cdot \nabla_\theta U_k(\theta). \tag{15}$$

We can rearrange the negative sign to group it with the utility gradient (since we want to maximize utility, $-\nabla U_k$ is the direction of descent for the loss):

$$\nabla_\theta \mathcal{L}_{\text{NBCG}} = \sum_{k=1}^{K} \underbrace{\left( \frac{\sigma(\Delta_k)}{\text{softplus}(\Delta_k) + \epsilon} \right)}_{\text{Exact Weight } \tilde{\alpha}_k(t)} \cdot \nabla_\theta(-U_k(\theta)). \tag{16}$$

## A.4. Approximation and Physical Interpretation

Equation 16 gives the exact numerical gradient. To understand the *mechanism* of NBCG (Eq. 7 in the main paper), we examine the behavior of the weight term $\tilde{\alpha}_k(t)$ under standard training conditions.

**Assumption:** We assume the coalition is operating in a valid bargaining region where the utility is at least marginally better than the disagreement point, i.e., $\Delta_k = U_k - d_k > 0$.

In this regime, we can apply the following approximations:

- As $x > 0$, $\text{softplus}(x) \approx x$.

- As $x > 0$, $\sigma(x) \approx 1$.

- The smoothing term $\epsilon$ is negligible (e.g., $10^{-8}$).

Applying these limits to $\tilde{\alpha}_k(t)$:

$$\lim_{\Delta_k \to 0^+} \tilde{\alpha}_k(t) = \frac{1}{\Delta_k} = \frac{1}{U_k(\theta) - d_k^{(t)}} \triangleq \alpha_k(t). \tag{17}$$

Thus, we recover the interpretable form presented in the main paper:

$$\nabla_\theta \mathcal{L}_{\text{NBCG}} \approx \sum_{k=1}^{K} \frac{1}{U_k(\theta) - d_k^{(t)}} \cdot \nabla_\theta(-U_k(\theta)). \tag{18}$$

## A.5. Analysis of Gradient Dynamics

The derived weight $\alpha_k(t) = \frac{1}{U_k - d_k}$ reveals why NBCG successfully breaks dominant coalition capture. We analyze the gradient magnitude $\|\nabla_\theta \mathcal{L}_k\|$ for two distinct scenarios:

**Case 1: Dominant Coalitions (Frequent Labels)** For coalitions dominated by frequent labels, the model quickly learns high-quality representations. Consequently, the current utility $U_k$ is significantly higher than the disagreement baseline $d_k$ (which is an exponential moving average and lags slightly behind).

$$U_k(\theta) \gg d_k^{(t)} \implies \Delta_k \text{ is Large} \implies \alpha_k(t) \to 0.$$

**Effect:** The gradient contribution from these well-learned coalitions is *suppressed*. This prevents the "majority vote" from dominating the shared encoder updates.

**Case 2: Under-served Coalitions (Rare Labels)** For rare label coalitions, the model struggles to improve utility. The utility $U_k$ hovers very close to the historical baseline $d_k$.

$$U_k(\theta) \approx d_k^{(t)} \implies \Delta_k \to 0 \implies \alpha_k(t) \to \infty.$$

**Effect:** As the "bargaining gap" narrows, the weight $\alpha_k$ spikes. This *amplifies* the gradient signal for rare labels, effectively demanding that the optimizer prioritize this coalition to satisfy the Nash condition.

**Pareto Efficiency:** Unlike static re-weighting (which fixes weights $w_k$ regardless of training state), $\alpha_k(t)$ is dynamic. Once a rare coalition's performance improves (increasing $\Delta_k$), its weight automatically decreases, shifting focus to other struggling coalitions. This dynamic equilibrium ensures a Pareto-efficient resource allocation.

## B. Dataset Details

In this section, we provide comprehensive statistics and a detailed distributional analysis of the four benchmark datasets used in our experiments: **Reuters-21578**, **Ohsumed**, **DBpedia**, and **20 Newsgroups**. These datasets were selected to represent diverse domains (news, medical, ontology) and varying degrees of label complexity and imbalance.

### B.1. Statistical Summary

Table 6 summarizes the key statistics for each dataset. We report the number of samples in train/validation/test splits, the total number of labels ($L$), the size of the vocabulary ($|V|$), and two critical multi-label metrics:

- **Label Cardinality:** The average number of labels associated with a single document ($\frac{1}{N} \sum_{i=1}^{N} |y_i|$).

- **Label Density:** The average number of labels per document divided by the total number of labels ($\frac{\text{Cardinality}}{L}$).

*Table 6.* Detailed statistics of the datasets used in this work. $L$ denotes the number of classes. Cardinality indicates the average number of labels per sample.

| Dataset | Domain | Labels ($L$) | Train | Val | Test | Avg. Length | Cardinality |
|---|---|---|---|---|---|---|---|
| **Reuters-21578** | News | 90 | 7,769 | 967 | 3,019 | 237 | 1.24 |
| **Ohsumed** | Medical | 23 | 3,357 | 750 | 4,000 | 165 | 1.63 |
| **DBpedia** | Ontology | 14 | 240,942 | 36,000 | 60,000 | 58 | 1.00 |
| **20 Newsgroups** | General | 20 | 11,314 | 1,884 | 7,532 | 392 | 1.00 |

*Note: "Avg. Length" refers to the average number of tokens per document after pre-processing.*

### B.2. Long-Tailed Label Distribution

A central motivation of NBCG is addressing the severe label imbalance inherent in these datasets. To visualize this challenge, we plot the label frequency distributions in Figure 7.

As observed, all datasets exhibit a characteristic **Power-Law (Zipfian) distribution**:

- **Head Labels:** A small fraction of labels (e.g., "earnings" in Reuters) accounts for the vast majority of training samples.

- **Tail Labels:** The majority of the label space consists of rare concepts (e.g., specific medical diseases in Ohsumed) that appear in fewer than 0.1% of the samples.

This structural imbalance justifies our definition of "Dominant Coalitions" (Head) and "Rare Coalitions" (Tail). Standard gradient descent tends to be biased towards the high-frequency Head labels, whereas NBCG's adaptive disagreement point mechanism allows the model to "negotiate" better representation for the Tail.

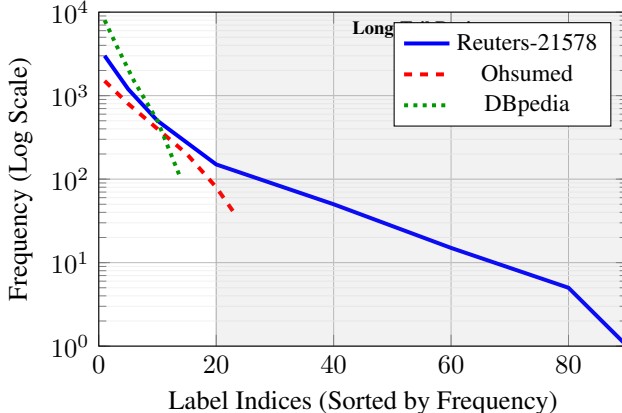

*Figure 7.* **Label Frequency Distribution (Log-Log Scale).** The datasets exhibit severe class imbalance. We sort labels by frequency (x-axis) and plot their occurrence counts (y-axis). The shaded region highlights the "Tail" labels, which are the primary focus of the NBCG optimization mechanism.

### B.3. Preprocessing

For all datasets, we followed standard preprocessing protocols consistent with prior work (e.g., TextGCN, HGAT):

1. **Tokenization:** We utilized the tokenizer from the pre-trained `RoBERTa-base` model.

2. **Length Truncation:** To accommodate GPU memory constraints while preserving semantic context, documents were truncated or padded to a fixed length (see Table 6 for average lengths).

3. **Label Filtering:** For Reuters-21578, we used the standard split where only classes with at least one training and one testing example are kept, resulting in 90 classes.

## C. Implementation Details

To ensure the reproducibility of our results, we provide a comprehensive description of the model architecture, training protocols, and the hardware/software environment used in our experiments.

### C.1. Model Architecture and Optimization

Our framework is implemented using PyTorch. The backbone encoder for all text inputs is **RoBERTa-base** (12 layers, 768 hidden size), initialized with pre-trained weights from HuggingFace. The training procedure operates in two synergistic phases as outlined in Algorithm 1.

**Phase 1: Structure Learning.** In this initial phase, we train the Neural Structural Equation Model (Neural SEM) to discover the adjacency matrix $A$. This process is lightweight and typically converges within 5 to 10 epochs. To ensure the learned structure is meaningful, we apply an L1 penalty ($\lambda_1$) to induce sparsity and a DAG constraint ($\lambda_{dag}$) to prevent cycles. The resulting matrix is then thresholded by $\tau$ to form the coalition partitions.

**Phase 2: Nash-Bargained Optimization.** Once coalitions are defined, we freeze the causal graph and proceed to train the shared encoder and label predictors. We employ the **AdamW** optimizer with a *discriminative learning rate strategy*:

the pre-trained RoBERTa encoder is fine-tuned with a lower learning rate (e.g., 1e-5) to preserve linguistic knowledge, while the randomly initialized label projection layers are trained with a higher learning rate (e.g., 1e-3). This prevents the "catastrophic forgetting" of pre-trained features.

## C.2. Hyperparameter Configuration

We performed a grid search on the validation set to determine the optimal hyperparameters. Table 7 details the specific settings for each dataset. Key parameters include the number of coalitions ($N$), the EMA decay rate for disagreement points ($\rho$), and the sparsity threshold ($\tau$). Note that $\rho$ is set to a high value (0.95–0.99) to ensure the disagreement point serves as a stable "status quo" reference rather than a volatile variable.

*Table 7.* **Hyperparameter Settings.** Optimal values across datasets. Constants: Optimizer is AdamW, Smooth $\epsilon = 1e - 8$.

| Param. | Reut. | Ohsu. | DBp. | 20Ng. |
|---|---|---|---|---|
| *General Training* | | | | |
| Batch / MaxLen | 16 / 256 | 16 / 256 | 32 / 128 | 16 / 512 |
| LR (Enc / Cls) | 1e-5 / 1e-3 | 1e-5 / 1e-3 | 2e-5 / 1e-3 | 1e-5 / 1e-3 |
| W. Decay | 1e-2 | 1e-2 | 1e-4 | 1e-2 |
| Epochs | 20 | 30 | 15 | 40 |
| *Phase 1 & 2: Model Specifics* | | | | |
| Sparsity $\lambda_1$ | 1e-4 | 1e-3 | 1e-4 | 1e-4 |
| DAG $\lambda_{\text{dag}}$ | 0.5 | 1.0 | 0.5 | 0.5 |
| Thresh. $\tau$ | 0.05 | 0.10 | 0.05 | 0.08 |
| Coalitions $N$ | 5 | 5 | 5 | 5 |
| EMA $\rho$ | 0.99 | 0.99 | 0.95 | 0.99 |

**Hardware and Software.** All experiments were conducted on a single server equipped with an **NVIDIA GeForce RTX 4090 (24GB VRAM)** GPU and an AMD EPYC 7003 CPU. The software environment is based on Python 3.9, PyTorch 2.1.0 (CUDA 11.8), and HuggingFace Transformers 4.35.0.

# D. Additional Experimental Results

This section complements the main text with a detailed breakdown of the ablation study and parameter sensitivity analysis.

## D.1. Full Ablation Study

In Section 4.6, we discussed the contribution of each module. Table 8 presents the complete numerical results across all four datasets. We compare the full NBCG model against three specific variants to isolate the source of performance gains.

First, the **w/o Nash (Weighted Sum)** variant replaces the Nash objective with a standard weighted sum of coalition losses. This configuration tests the core game-theoretic assumption; as shown in the table, it consistently suffers the largest performance drop (e.g., -4.2% on Reuters), confirming that the dynamic negotiation mechanism is superior to static aggregation.

Second, the **w/o Neural SEM (Random)** variant replaces the learned causal structure with random label partitioning. The performance degradation observed here (approx. 2-3%) validates that "who plays with whom" matters—grouping causally related labels maximizes the efficiency of gradient sharing.

Finally, **w/o Adaptive $d_k$ (Fixed)** fixes the disagreement point at zero. The drop in performance indicates that the moving "status quo" baseline is essential for adaptively identifying which coalitions are currently under-served.

## D.2. Parameter Sensitivity: EMA Rate $\rho$

The Exponential Moving Average (EMA) rate $\rho$ controls the stability of the disagreement point $d_k$. Table 9 illustrates the trade-off on DBpedia. A value of $\rho = 1.0$ effectively freezes the baseline, failing to adapt to the model's learning progress. Conversely, a very low $\rho$ (e.g., 0.5) makes the baseline too volatile, causing optimization instability. We observe that $\rho \in [0.95, 0.99]$ yields the most robust performance.

=

*Table 8.* **Ablation Study.** Rare-F1 @ 30%. $\Delta$ denotes performance drop.

| Variant | Reut. | Ohsu. | DBp. | 20Ns. |
|---|---|---|---|---|
| **Full NBCG** | **65.5** | **66.1** | **65.8** | **78.5** |
| − Nash Game | 61.3 (↓4.2) | 61.5 (↓4.6) | 61.8 (↓4.0) | 74.2 (↓4.3) |
| − Neural SEM | 63.8 (↓1.7) | 63.0 (↓3.1) | 63.5 (↓2.3) | 76.1 (↓2.4) |
| − Adapt. $d_k$ | 64.2 (↓1.3) | 64.8 (↓1.3) | 64.2 (↓1.6) | 77.0 (↓1.5) |

*Table 9.* **Sensitivity Analysis of EMA Rate $\rho$ (DBpedia).**

| EMA $\rho$ | 0.5 | 0.8 | 0.9 | **0.95** | 0.99 |
|---|---|---|---|---|---|
| R-F1 (30%) | 63.1 | 64.5 | 65.2 | **65.8** | 65.4 |
| mAP | 88.2 | 89.1 | 89.5 | **89.8** | 89.6 |

# E. Baseline Reproducibility

We benchmark NBCG against a wide range of state-of-the-art methods. To ensure a fair comparison, we standardized the text encoder backbone and data splits across all baselines.

**RoBERTa (Base).**   We fine-tuned the `roberta-base` model using a standard Binary Cross Entropy (BCE) loss. This serves as the foundation for all other methods to isolate the architectural gains from the pre-training gains.

**Loss-Reweighting Methods.**   We implemented **Focal Loss**, **Class-Balanced (CB) Loss**, and **DB-Loss** on top of the RoBERTa backbone. The class-balanced weights were strictly calculated based on the training set statistics using the formula $w_j \propto (1 - \beta)/(1 - \beta^{n_j})$, with $\beta$ searched in $\{0.9, 0.99, 0.999\}$.

**Graph-Based Methods.**   For **HGAT**, **TextGCN**, and **HyperGAT**, we utilized the official implementations provided by the respective authors. A critical modification was made for fairness: since these methods originally relied on LSTM or static GCN encoders, we adapted them to accept RoBERTa embeddings as node features. This ensures that performance differences stem from the graph reasoning capabilities rather than the encoder strength.

**CCG ((Fan et al., 2025)).**   As the most recent and strongest baseline, we used the official code of CCG. We ensured that the causal graph construction step in CCG used the exact same pre-processed data as NBCG to guarantee a direct comparison of the game-theoretic mechanisms.

All baselines underwent a hyperparameter search (Learning Rate $\in \{1e-5, 2e-5, 5e-5\}$, Batch Size $\in \{16, 32\}$) and we report the best test performance derived from the checkpoint with the highest validation mAP.

