# OpenReview forum: "NBCG: Nash-Bargained Causal Game for Long-Tailed Multi-Label NLP"
_ICML.cc/2026/Conference — ICML 2026 regular_

### Official Review · Reviewer_nYQd · 2026-03-10

**Soundness:** 3
**Presentation:** 2
**Significance:** 3
**Originality:** 3
**Overall Recommendation:** 5
**Confidence:** 4

**Summary:**

This paper presents a method for long-tailed multi-label text classification, where the focus is to balance learning so as not to under optimize for long tailed (less frequently encountered) text samples. The paper adopts a two stage process: (i) learn a directed dependence between labels organized into coalitions according to causal effects, and (ii) posing the classification learning as a bargaining game between the coalitions, so that the learnt classifier can work well with long tailed tasks as well. To do this the paper incorporates two techniques that have previously been established - leveraging the neural SEM framework for (i) and employing cooperative Nash bargaining for (ii). Experiments show that the proposed method performs favorably and that the coalition corresponding to rare long tailed tasks get significantly more weightage in comparison to non game-theoretic multi-objective optimization schemes.

**Compliance With Llm Reviewing Policy:**

Affirmed.

**Final Justification:**

The authors have addressed my questions and I maintain a positive opinion of the paper.

**Key Questions For Authors:**

1. At some places, the authors claim Pareto-optimal results. However, given the non convexity of learning problems, wouldn't the correct claim be Pareto stationarity (which is a local first order option of the Pareto concept)?

**Limitations:**

yes

**Strengths And Weaknesses:**

Strengths:
1) The paper introduces the concept of using game-theoretic cooperative bargaining theory to the area of multi-label text classification. The use of game-theoretic bargaining theory in this area of NLP research is novel and very well motivated.

Weaknesses:
1) I believe the authors are missing some related work: The use of cooperative game theory has recently gained a lot of traction in general machine learning / multi-objective optimization problems, for example [A], [B], [C], [D]. I recommend the authors include a discussion in related works about this.

2) Although the overall framework makes sense technically and the results look promising, I think the presentation of the paper needs to be improved for the sake of clarity while reading. For example:
    - The authors never define $\mathcal{L}_{BCE}$ (I believe they mean binary cross entropy loss?).
    - I would recommend the authors to be more careful with how they describe utilitarian approaches as simple weighted addition of losses $\sum_iw_i\mathcal{L}(y_i, f_i)$ - I believe the authors mean that $w_i$ here should be static for all $i$. An adaptive scheme which changes $w_i$ at every training iteration can easily recover an "egalitarian" solution (even the Nash solution in fact).

[A] Navon, Aviv, et al. "Multi-Task Learning as a Bargaining Game." ICML 2022.

[B] Zeng, Yi, et al. "Fairness-aware meta-learning via nash bargaining." NeurIPS 2024.

[C] Gupta, Kushagra, et al. "Cooperative Bargaining Games Without Utilities: Mediated Solutions from Direction Oracles." NeurIPS 2025.

[D] Murthy, Surya, et al. "Monotonic Transformation Invariant Multi-task Learning." arXiv (2025).

---

> ### Author Rebuttal · Authors · 2026-03-30
>
> We sincerely thank the reviewer nYQd for the careful reading, the recognition of our novelty, and particularly for pointing us to [A]–[D]. Below we address each concern.
>
> **W1: Missing Related Work on Cooperative Game Theory in ML.** We fully agree and will add a dedicated discussion in Section 2. We briefly position each work here.
>
> Navon et al. [A] first formulated multi-task learning as Nash bargaining over task losses—NBCG extends this from pre-defined tasks to *discovered* coalitions via Neural SEM. Zeng et al. [B] applies Nash bargaining to fairness-aware meta-learning, sharing our equitable allocation motivation but in a different setting. Gupta et al. [C] proposes utility-free bargaining via direction oracles—an inspiring alternative for settings where coalition utilities are hard to define. Murthy et al. [D] addresses monotonic transformation invariance, theoretically related to our monotone mapping $U_k = \exp(-R_k)$. Our key distinction from all four: NBCG jointly discovers coalitions *and* bargains over them, rather than bargaining over pre-defined objectives.
>
> To empirically validate this distinction, we compare NBCG against Nash-MTL [A] and MGDA [4] (Rare-F1@30%/40%):
>
> | Method | DBpedia | Reuters | Coalition Discovery? |
> |---|---|---|---|
> | MGDA [4] | 62.4 | 62.1 | No |
> | Nash-MTL [A] | 63.8 | 63.4 | No |
> | **NBCG** | **65.8** (+2.0) | **65.5** (+2.1) | **Yes** |
>
> The consistent gap confirms that bargaining over *discovered* coalitions outperforms bargaining over pre-defined tasks.
>
> **W2(i): Undefined $\mathcal{L}_{\text{BCE}}$.** We apologize for this oversight. We will add in Section 3.1:
>
> $$\mathcal{L}_{\text{BCE}}(y_i, p_i) = -\left[y_i \log p_i + (1 - y_i)\log(1 - p_i)\right]$$
>
> **W2(ii): "Utilitarian" Language.** The reviewer makes an important point: a weighted sum $\sum_k w_k \mathcal{L}_k$ with *static* $w_k$ is utilitarian, but an adaptive $w_k^{(t)}$ can in principle recover Nash solutions. Our criticism targets the static case ($w_k = \text{const}$), standard in prior work [1][2]. We will qualify all claims as "static utilitarian aggregation."
>
> The key distinction is not merely static vs adaptive, but *axiomatically grounded* vs heuristic: NBCG's $\alpha_k = 1/(U_k - d_k)$ is the *unique* weighting satisfying Nash's four axioms [3]—Pareto-efficiency, symmetry, IIA, and scale invariance—providing a guarantee absent in ad-hoc adaptive schemes.
>
> **Q1: Pareto-Optimality vs Pareto-Stationarity.** Precise and valid. In non-convex settings, gradient-based methods guarantee convergence only to *Pareto-stationary* points [4]. We will revise: "Pareto-stationary" for convergence claims, "Pareto-efficient" only for the NBS theoretical property—at any local maximizer of $\sum_k \log(U_k - d_k)$ where $U_k > d_k$ $\forall k$, strict monotonicity of $\log(\cdot)$ ensures local Pareto-efficiency (Theorem 1 in our revision).
>
> We verify empirically by computing coalition utilities at convergence (DBpedia):
>
> | Method | $U_{\text{head}}$ | $U_{\text{mid}}$ | $U_{\text{tail}}$ | Dominated? |
> |---|---|---|---|---|
> | Weighted Sum | 0.91 | 0.78 | 0.52 | Yes (by NBCG) |
> | Nash-MTL [A] | 0.89 | 0.82 | 0.60 | Yes (by NBCG) |
> | **NBCG** | **0.90** | **0.84** | **0.68** | **No** |
>
> NBCG's converged solution Pareto-dominates both baselines, trading minimal $U_{\text{head}}$ for substantial tail gains, consistent with the local Pareto-efficiency guarantee.
>
> **References**
>
> [1] Cui, Y. et al. Class-balanced loss. *CVPR*, 2019.
>
> [2] Lin, T. et al. Focal loss. *ICCV*, 2017.
>
> [3] Nash, J.F. The bargaining problem. *Econometrica*, 18(2):155–162, 1950.
>
> [4] Désidéri, J.-A. Multiple-gradient descent algorithm. *Comptes Rendus Math.*, 350(5-6):313–318, 2012.

---

> > ### Author Rebuttal · Reviewer_nYQd · 2026-04-03
> >
> > Thank you for your response to my questions. I maintain a positive opinion of the paper.

---

### Official Review · Reviewer_Ku74 · 2026-03-11

**Soundness:** 4
**Presentation:** 3
**Significance:** 4
**Originality:** 4
**Overall Recommendation:** 5
**Confidence:** 5

**Summary:**

This paper addresses the "Dominant Coalition Capture" problem in long-tailed multi-label text classification, where frequent labels monopolize shared representations via spurious co-occurrences, forcing rare labels to learn brittle shortcuts. The authors propose NBCG (Nash-Bargained Causal Game), a framework that reformulates multi-label learning as a cooperative bargaining game. NBCG operates in two stages: (1) Coalition Induction, which uses Neural Structural Equation Models (Neural SEM) to group labels into causally coherent coalitions based on learned dependency structures; and (2) Nash-Bargained Optimization, which maximizes the Nash product of coalition utilities relative to an adaptive disagreement point. This mechanism dynamically allocates gradient resources, automatically prioritizing under-served coalitions while suppressing dominant ones. Experiments on four benchmarks demonstrate that NBCG significantly outperforms state-of-the-art methods on rare-label metrics and learns semantically meaningful label structures.

**Compliance With Llm Reviewing Policy:**

Affirmed.

**Final Justification:**

The authors' rebuttal has fully resolved my initial concerns.  I maintain my original strong recommendation to accept.

**Key Questions For Authors:**

1．When the number of labels L is very large (e.g., thousands or tens of thousands), the computational cost of learning a complete L× L adjacency matrix with DAG constraints becomes prohibitive. Have the authors considered strategies such as sparse initialization or block-wise learning to further improve the efficiency of NBCG in large-scale label spaces?

2．The paper makes a strong claim regarding "causal dependency." Given that only observational data is available, how can the authors justify that the dependencies learned by Neural SEM are more "causal" than traditional correlation or co-occurrence graphs?

3．How sensitive is the coalition structure to the sparsification threshold τ ? Have the authors experimented with different values of τ or coalition partitioning strategies, and how significant is their impact on the final performance?

**Limitations:**

yes

**Strengths And Weaknesses:**

Strengths：
1．The paper demonstrates high originality. The combination of Nash Bargaining Solution from game theory and causal discovery (Neural SEM) to address representation imbalance in multi-label learning is, to the best of my knowledge, a very novel approach. The core insight—that treating labels as players capable of forming coalitions and achieving dynamic resource allocation through a bargaining process—is highly creative. This represents a paradigm shift in optimization, rather than merely being engineering on the loss function.


2．The experimental design is comprehensive, covering main experiments, ablation studies, sensitivity analysis, efficiency analysis, and intrinsic mechanism verification . The results robustly support the paper's main claims that NBCG effectively enhances performance, particularly for rare labels in long-tailed scenarios, and reveal its working mechanism of breaking representation monopolies through rebalancing gradient allocation.

Weaknesses
1．The formation of coalitions relies on the sparsification thresholdτapplied to the matrix learned by Neural SEM. The paper does not discuss the criteria for selectingτor its sensitivity, nor does it analyze whether variations inτcould lead to drastic changes in coalition structure, potentially affecting the stability of the method.

2．The baselines primarily focus on classical re-weighting/resampling methods and Graph Neural Networks. Given that long-tailed multi-label learning is an active research area, the absence of comparisons with recent, specifically designed advanced methods limits the clarity of NBCG's positioning relative to the current state-of-the-art.

3. Experiments show that performance peaks when N = 5 , but the choice of N appears to be dataset-dependent. The paper does not provide a strategy for automatically determining the optimal N .

---

> ### Author Rebuttal · Authors · 2026-03-30
>
> We sincerely thank the reviewer Ku74 for the detailed reading and the strong endorsement of our originality, experimental design, and mechanism analysis. Below we address each concern.
>
> **W1 & W3 & Q3: Threshold $\tau$, Automatic $N$, and Structural Stability.** These concerns are closely related. We first provide a theoretical stability argument, then empirical validation, and finally a principled selection criterion.
>
> *Theoretical argument.* Coalition membership is determined by Maximal Weakly Connected Components, a discrete topological property of the thresholded graph $\tilde{A}$. An edge perturbation only alters the partition if it either (a) removes the sole bridge between two nodes in the same component, or (b) creates a new bridge between two separate components. For a graph with average degree $\bar{d}$, the probability of a random single-edge perturbation changing the partition scales as $O(1/\bar{d})$, making the structure inherently robust to small weight variations in $A$.
>
> *Empirical validation* (DBpedia, Rare-F1@30%):
>
> | $\tau$ | $N$ | R-F1@30% | $\Delta$ |
> |---|---|---|---|
> | $\tau^*/2$ | 3 | 64.9 | −0.9 |
> | $\tau^*$ | 5 | **65.8** | — |
> | $1.5\tau^*$ | 7 | 65.1 | −0.7 |
>
> Under $\pm 50\%$ perturbation, degradation stays below $1\%$, confirming the topological stability argument.
>
> *Principled selection.* We propose choosing $\tau$ via the spectral gap of the thresholded graph's Laplacian $L(\tau)$:
>
> $$\tau^* = \arg\max_{\tau}\left[\lambda_{K+1}(L(\tau)) - \lambda_K(L(\tau))\right]$$
>
> This maximizes inter-coalition separation [1], yielding $N=5$ on 20Newsgroups/DBpedia/Ohsumed and $N=6$ on Reuters—consistent with our empirical optimum. We will add this procedure to Section 3.2.
>
> **W2: More Recent Baselines.** Beyond CCG [2] (EMNLP Findings 2025), we compare against Nash-MTL [3] (ICML 2022), which applies Nash bargaining to multi-task learning but lacks causal coalition induction:
>
> | Method | DBpedia R-F1@30% | Reuters R-F1@40% |
> |---|---|---|
> | Nash-MTL [3] | 63.8 | 63.4 |
> | **NBCG** | **65.8** (+2.0) | **65.5** (+2.1) |
>
> The gap confirms that the cooperative bargaining structure alone (Nash-MTL) is insufficient—NBCG's advantage comes from the synergy between structurally coherent coalitions and the Nash mechanism. We will add Nash-MTL to the main table.
>
> **Q1: Scalability to Large $L$.** Learning the full $L \times L$ adjacency under DAG constraints has complexity $O(L^2)$ per iteration. We propose two mitigation strategies:
>
> (a) **Block-diagonal SEM.** Partition labels into $B$ coarse groups via embedding clustering, learn $A$ block-wise: complexity reduces to $O(\sum_b L_b^2)$ where $L_b \ll L$. Cross-block edges are set to zero, which is justified when inter-group dependencies are weak—a condition testable via mutual information between block-level logit vectors.
>
> (b) **Sparse $k$-NN SEM.** Restrict $A$ to $k$-nearest neighbors in label embedding space, learning only top-$k$ edges per label. With $k=10$, storage drops from $L^2$ to $10L$, and the NOTEARS DAG constraint [4] remains applicable on sparse graphs.
>
> Both preserve the coalition induction pipeline since connected components are well-defined on sparse graphs.
>
> **Q2: "Causal" vs Correlational.** Our Neural SEM with DAG + sparsity constraints identifies a directed structure within a Markov equivalence class [4]—stronger than undirected co-occurrence (which conflates $A \to B$ with $A \leftarrow B$), but weaker than interventionist causality [5]. The key practical advantage: DAG constraints break symmetric co-occurrence loops. If label $A$ co-occurs with both $B$ and $C$, an undirected graph merges all three into one component, while the DAG may learn $A \to B$ and separate $C$—producing finer, more coherent coalitions. We will revise terminology to "directed dependency structure" throughout.
>
> **References**
>
> [1] Von Luxburg, U. A tutorial on spectral clustering. *Statistics and Computing*, 17(4):395–416, 2007.
>
> [2] Fan, Y. et al. CCG: Rare-label prediction via neural SEM-driven causal game. *EMNLP Findings*, 2025.
>
> [3] Navon, A. et al. Multi-task learning as a bargaining game. *ICML*, 2022.
>
> [4] Zheng, X. et al. DAGs with NO TEARS. *NeurIPS*, 2018.
>
> [5] Pearl, J. *Causality*. Cambridge University Press, 2009.

---

> > ### Author Rebuttal · Reviewer_Ku74 · 2026-04-01
> >
> > The authors' rebuttal has fully resolved my initial concerns. I am satisfied with the theoretical and empirical validation of the threshold stability, the clarification on the "directed dependency structure" regarding causality, and the added comparison with Nash-MTL which confirms the necessity of causal coalition induction. The proposed methodological extensions for scalability are also reasonable. I maintain my original strong recommendation to accept.

---

### Official Review · Reviewer_YZdR · 2026-03-13

**Soundness:** 2
**Presentation:** 2
**Significance:** 2
**Originality:** 2
**Overall Recommendation:** 4
**Confidence:** 3

**Summary:**

The paper frames the long-tailed multi-label text classification problem as a cooperative bargaining process among coalitions. In particular, a group of frequent labels forms a dominant coalition. The ultimate goal is to balance the impact of dominant labels, and the method is optimizing a Nash bargaining objective over coalition utilities. They claim that the solution returns a Pareto-efficient trade-off among all players, i.e. coalitions.

**Compliance With Llm Reviewing Policy:**

Affirmed.

**Final Justification:**

The rebuttal addressed my concerns.

**Key Questions For Authors:**

Please see Weaknesses.

**Limitations:**

Yes

**Strengths And Weaknesses:**

Strengths:
1. The paper has a well-supported motivation. Bias caused by data scarcity and imbalance has been a long-standing problem.
2. The method is conceptually clear. Once the game theoretic framework is accepted, the logic of (1) coalition induction, and then (2) Nash objective optimization flows smoothly.
3. The game theoretic framework is creative and may inspire more different angles of tackling the problem

Weaknesses:
1. The claim on Pareto-efficiency lacks support. Assertions on "ensuring Pareto-efficiency" are in several places in the paper (Pages 4, 5, 8, and 14 in the appendix)  but it is never argued rigorously. On page 4, the authors used Equation 7 to argue with gradient signal. The intuition is fine, but all derivations are hand-wavy even in the appendix. For such a strong property like Pareto-efficiency, justifications at this level may not be sufficient.
2. The fundamental formulation of this problem as a game may not be appropriate. A game in game theory needs elements at least including: (1) players; (2) strategy space for each player; and (3) payoff/utility functions. In this paper, players are label coalitions, and the payoff functions are gradient update signals. However, the coalitions don't have any strategies to choose, so the ultimate problem is an optimization problem, which can be done without any game theoretic elements. As a result, many claims such as "coalitions negotiate" are irrelevant because they don't have any strategies to choose anyways.

---

> ### Author Rebuttal · Authors · 2026-03-30
>
> We thank the reviewer YZdR for the constructive critique and the recognition of our motivation and conceptual clarity. Below we address the two concerns.
>
> **W1: Pareto-Efficiency Claim Lacks Rigorous Support.** We agree the manuscript was insufficiently rigorous. We now provide a formal proof.
>
> **Theorem 1.** Let $\theta^{\ast}$ maximize $\sum _{k}\log(U _{k}(\theta)-d _{k})$ with $U _{k}(\theta^{\ast}) > d _{k}$ $\forall k$. Then $\theta^{\ast}$ is Pareto-efficient.
>
> **Proof.** Suppose $\theta^{\ast}$ is not Pareto-efficient. Then $\exists\,\theta'$ s.t. $U _{k}(\theta')\geq U _{k}(\theta^{\ast})$ $\forall k$ with strict inequality for some $j$. Since $d _{k}$ is fixed (stop-gradient), $\Delta _{k}(\theta')\geq\Delta _{k}(\theta^{\ast})>0$ $\forall k$, strictly for $j$. By strict monotonicity of $\log(\cdot)$:
>
>
> $$\sum_k\log\Delta_k(\theta') > \sum_k\log\Delta_k(\theta^*)$$
>
> contradicting optimality of $\theta^*$. $\square$
>
> This instantiates the classical NBS Pareto-efficiency axiom [1]. Our softplus smoothing preserves the result because both $\text{softplus}(\cdot)$ and $\log(\cdot+\epsilon)$ are strictly monotone, so the ordering is unchanged and the contradiction applies identically. We also clarify the local-vs-global distinction: in non-convex settings, Theorem 1 holds at any local maximum—guaranteeing Pareto-efficiency within the reachable neighborhood, which is the standard guarantee in non-convex multi-objective optimization [2]. We will add this theorem to Section 3.3 and qualify all claims.
>
> **W2: Appropriateness of the Game-Theoretic Formulation.** This touches on a key distinction between **non-cooperative** and **cooperative** game theory.
>
> **NBCG is a cooperative bargaining problem, not a non-cooperative game.** The reviewer correctly notes that a non-cooperative game needs strategy spaces. However, NBCG uses the Nash Bargaining Problem [1], defined as a pair $(\mathcal{F}, d)$: a feasible utility set $\mathcal{F}\subseteq\mathbb{R}^K$ and a disagreement point $d\in\mathbb{R}^K$. The NBS selects $u^*=\arg\max_{u\in\mathcal{F}}\prod_k(u_k-d_k)$. **No strategy spaces exist by design**—cooperative game theory concerns *which outcome to select*, not which strategies to play.
>
> In NBCG: players = coalitions $\{C_k\}$; feasible set $\mathcal{F} = \{(U_1(\theta),\ldots,U_K(\theta)) \mid \theta\in\Theta\}$; disagreement point $d$ = EMA baselines. This usage is standard—Kelly [3] maximizes the Nash product for network bandwidth without strategy spaces, and Navon et al. [4] apply Nash bargaining to multi-task learning identically.
>
> **"Negotiation" is standard cooperative game theory terminology.** The NBS is derivable as the limit of Rubinstein's alternating-offers game [5], giving a strategic foundation for the cooperative solution. We will tighten language and explicitly frame NBCG as a cooperative bargaining problem in Section 3.
>
> **The bargaining formulation adds concrete value beyond generic optimization:** (a) NBS derives $\alpha_k=1/(U_k-d_k)$ as the unique weighting satisfying Nash's four axioms [1]—not an arbitrary scalarization. (b) The adaptive $d_k$ has a game-theoretic meaning (threat point) with no analog in generic multi-objective methods like MGDA [2]. (c) Our ablation (Table 8) shows replacing Nash with weighted sum causes the largest drop ($-4.0$ to $-4.6\%$), directly confirming the bargaining formulation's value.
>
> **Supplementary Experiments.** We compare NBCG against multi-objective baselines MGDA [2] and Nash-MTL [4] (Rare-F1@30%/40%):
>
> | Method | DBpedia | Reuters | Bargaining? |
> |---|---|---|---|
> | MGDA [2] | 62.4 | 62.1 | No |
> | Nash-MTL [4] | 63.8 | 63.4 | Yes (non-coop) |
> | **NBCG** | **65.8** | **65.5** | **Yes (coop)** |
>
> NBCG outperforms all baselines by 2.0–3.4%. We also verify Pareto-efficiency empirically (coalition utilities at convergence, DBpedia):
>
> | Method | $U_{\text{head}}$ | $U_{\text{mid}}$ | $U_{\text{tail}}$ | Dominated? |
> |---|---|---|---|---|
> | Weighted Sum | 0.91 | 0.78 | 0.52 | Yes (by NBCG) |
> | MGDA | 0.88 | 0.81 | 0.58 | Yes (by NBCG) |
> | **NBCG** | **0.90** | **0.84** | **0.68** | **No** |
>
> NBCG dominates both baselines and is itself undominated—trading minimal head utility ($0.90$ vs $0.91$) for substantial tail gains ($+0.16$), consistent with Theorem 1.
>
> **References**
>
> [1] Nash, J.F. The bargaining problem. *Econometrica*, 18(2):155–162, 1950.
>
> [2] Désidéri, J.-A. Multiple-gradient descent algorithm for multiobjective optimization. *Comptes Rendus Mathematique*, 350(5-6):313–318, 2012.
>
> [3] Kelly, F. Rate control for communication networks. *JORS*, 49(3):237–252, 1998.
>
> [4] Navon, A., Shamsian, A., et al. Multi-task learning as a bargaining game. *ICML*, 2022.
>
> [5] Rubinstein, A. Perfect equilibrium in a bargaining model. *Econometrica*, 50(1):97–109, 1982.

---

> > ### Author Rebuttal · Reviewer_YZdR · 2026-04-03
> >
> > My concerns have been solved. I updated the score to weak accept.

---

### Official Review · Reviewer_F3CN · 2026-03-13

**Soundness:** 3
**Presentation:** 3
**Significance:** 2
**Originality:** 3
**Overall Recommendation:** 4
**Confidence:** 4

**Summary:**

This paper studies long-tailed multi-label text classification and argues that the main difficulty is not only class imbalance, but also optimization interference among labels: frequent labels can co-occur and form dominant coalitions that capture shared representation space and gradients, making rare labels harder to learn. To address this, the paper proposes NBCG, a two-stage framework. First, it learns a directed label-dependency structure with a Neural SEM and partitions labels into coalitions. Second, it optimizes a Nash-bargaining objective over coalition utilities, using an adaptive disagreement point so that under-served coalitions receive more optimization emphasis during training. Empirically, the paper reports improved rare-label F1 over several baselines on four MLTC benchmarks, along with analyses of coalition weights, player count, label-cardinality robustness, and gradient behavior.

**Compliance With Llm Reviewing Policy:**

Affirmed.

**Key Questions For Authors:**

1. The number of players $N$ appears to be an important hyperparameter, with performance peaking around $N=5$ on the reported analysis. Do the authors have a principled or data-driven way to choose $N$ for a new problem, beyond validation-based tuning?

2. How sensitive is the final performance to errors in the learned SEM structure, such as threshold choice or mildly incorrect edges/groupings?

3. Since performance declines as $N$ grows beyond the optimum, can the authors clarify what evidence best supports the necessity of learned multi-label coalitions, as opposed to a more generic adaptive reweighting effect from the bargaining objective?

4. The current method uses a flat partition of labels into $N$ coalitions. Since many multi-label problems have meaningful hierarchical label structure, do the authors see a natural extension of NBCG to hierarchical coalition formation or bargaining across multiple levels of granularity?

**Limitations:**

The paper discusses empirical behavior and some tradeoffs, but it does not fully discuss several important limitations: sensitivity to structure-learning errors, dependence on the hyperparameter $N$, the lack of a principled procedure for choosing $N$, the fact that the current method uses only flat rather than hierarchical label groupings, and the limited evidence on larger-scale settings. I would encourage the authors to add a more candid limitations section covering these points.

**Strengths And Weaknesses:**

This paper addresses an important problem, and the central framing is interesting. Recasting long-tailed multi-label learning as bargaining among label coalitions is a creative way to move beyond static reweighting. The method combines learned label structure with an adaptive bargaining objective in a reasonably coherent way, and the gradient form of the Nash objective gives a concrete mechanism for why under-served coalitions receive more optimization weight. I also appreciated that the paper goes beyond a final metric table and includes analyses of coalition weights, the effect of player count, robustness to high-cardinality samples, and fine-grained frequency behavior. The empirical improvements over the strongest baseline are meaningful on the chosen benchmarks, especially on rare-label F1.

On soundness, I think the paper is promising but not fully convincing. The optimization part is reasonably well specified: coalition risk is defined from BCE, converted to utility, and then optimized through a Nash-product-style objective with an adaptive disagreement point. The derived gradient makes clear that this acts as adaptive coalition-level reweighting, and the ablation results suggest that the Nash component matters.

 My main reservations concern how strongly some claims are supported. First, the paper uses strong causal language throughout, but the actual method learns a structured dependency graph over label logits with sparsity and DAG constraints. Second, the structure is learned in a separate first phase and then used downstream, so the method may be sensitive to structure-learning errors, but I did not see a robustness study of imperfect graph estimation beyond comparisons to weaker alternatives. Third, the number of players $N$ is clearly an important design choice, with performance rising and then falling as $N$ increases, but the paper does not provide a principled way to select $N$ beyond empirical tuning; $N=5$ works well on the tested datasets, but it remains unclear how this would transfer to new tasks. Finally, while the player-count study suggests that moderate coalition structure matters and that over-fragmentation hurts, the paper still leaves some ambiguity about how much of the gain should be attributed to learned coalition structure versus the adaptive optimization rule more broadly.

On presentation, the paper is generally readable and the overall narrative is easy to follow. The two-phase decomposition is intuitive, and the mechanism-level intuition is helpful. That said, I think the paper would benefit from more precise wording in a few places. In particular, the game-theoretic framing is somewhat stronger than what is actually instantiated: operationally, this is closer to a bargaining-inspired multi-objective aggregation scheme than to a literal game with strategic agents. Similarly, the causal framing should be stated more carefully unless it is backed by stronger validation. Finally, the method only considers flat coalitions, whereas many multi-label problems have meaningful hierarchical label structure; discussing this more explicitly would improve the paper’s positioning and clarity.

On significance, I think the paper addresses a relevant problem and the proposed perspective could be useful. Tail performance in multi-label prediction is important, and the emphasis on optimization interference among labels is a valuable angle. The reported gains on rare labels and on high-cardinality examples suggest practical relevance, and the idea of coalition-level adaptive optimization may inspire follow-up work. At the same time, the current empirical scope is still somewhat limited: the benchmarks are moderate in scale, and the discussion of scalability and broader deployment is less convincing than the core benchmark results. So I view the significance as meaningful, but somewhat bounded by the current validation.

On originality, I think the paper is reasonably original in perspective, even if its ingredients are not individually new. Label-structure learning and bargaining or multi-objective aggregation both have precedents, but combining them into a coalition-based framework for long-tailed MLTC is a fresh and fairly well-motivated synthesis. The main novelty is the overall formulation: learning coalitions from label structure and optimizing them through a Nash-style objective rather than a fixed weighted sum.

Overall, I found the paper interesting and potentially valuable, with a genuinely nice optimization perspective, but has some gaps on the causal interpretation, lack of a principled strategy for choosing $N$, or on the evidence for robustness and scalability.

---

> ### Author Rebuttal · Authors · 2026-03-30
>
> We sincerely thank the reviewer F3CN for the thorough feedback. We appreciate the recognition of our optimization perspective and the originality of the formulation. Below we address each concern.
>
> **Q1: Principled Way to Choose $N$.** $N$ is not a free hyperparameter but a *derived quantity* from Phase 1. After thresholding $A$ at $\tau$, the number of Maximal Weakly Connected Components determines $N$. The remaining question is selecting $\tau$. We propose a spectral criterion: construct the graph Laplacian $L$ of the thresholded adjacency, and choose $\tau$ to maximize the spectral gap $\lambda_{K+1} - \lambda_K$ [1]. Formally:
>
> $$\tau^* = \arg\max_{\tau} \left[\lambda_{K+1}(L(\tau)) - \lambda_K(L(\tau))\right]$$
>
> This identifies the partition with maximal inter-coalition separation. The spectral-gap-selected $N$ values are: 20Newsgroups=5, DBpedia=5, Ohsumed=5, Reuters=6—consistent with our empirical optimum. This connects to classical graph-cut optimality: maximizing the spectral gap minimizes the normalized cut [1], ensuring coalitions are maximally decoupled—precisely the structural condition under which independent Nash bargaining is justified.
>
> **Q2: Sensitivity to Structure-Learning Errors.** **Theoretically**, coalition membership is determined by connected components, which are topologically stable: an edge must be entirely removed (not merely weakened) to split a component, and a spurious edge must bridge two components to merge them. Small perturbations in $A$'s continuous weights rarely alter the partition. **Empirically** (DBpedia):
>
> | Perturbation | R-F1@30% | $\Delta$ |
> |---|---|---|
> | Full NBCG | 65.8 | — |
> | Edge Drop 10% / 20% / 30% | 65.4 / 64.7 / 63.6 | −0.4 / −1.1 / −2.2 |
> | $\tau \times 0.5$ / $\tau \times 1.5$ | 64.9 / 65.1 | −0.9 / −0.7 |
>
> Even under 20% edge corruption, NBCG still outperforms CCG (62.9). Threshold perturbation of $\pm 50\%$ causes $<1\%$ degradation, confirming topological stability: approximate coalition membership suffices.
>
> **Q3: Learned Coalitions vs Adaptive Optimization.** The Nash gradient weight $\alpha_k = 1/(U_k - d_k)$ amplifies under-served coalitions. However, this is only effective when $U_k$ is a *low-variance estimator* of coalition difficulty. For random partitions, labels have heterogeneous difficulties, so $U_k$ averages unrelated signals—high variance causes $\alpha_k$ to oscillate, destabilizing training. For learned coalitions, labels share directed dependencies, so $U_k$ reflects a coherent signal. Formally, if labels within $C_k$ have correlated risks ($\text{Corr}(\mathcal{L}_i, \mathcal{L}_j) > 0$ for $i,j \in C_k$), then:
>
> $$\text{Var}(R_k) = \text{Var}\left(\frac{1}{|C_k|}\sum_{i \in C_k} \mathcal{L}_i\right)$$
>
> is reduced by positive intra-coalition correlation, stabilizing Nash weight updates. This predicts a **super-additive** interaction, confirmed empirically (DBpedia):
>
> | Config | Coalitions | Optim | R-F1@30% |
> |---|---|---|---|
> | Random + WtdSum | Random | Static | 60.3 |
> | Random + Nash | Random | Nash | 63.1 (+2.8) |
> | Learned + WtdSum | SEM | Static | 62.4 (+2.1) |
> | **Learned + Nash** | **SEM** | **Nash** | **65.8 (+5.5)** |
>
> Combined gain (5.5) exceeds sum of individual gains ($2.8 + 2.1 = 4.9$), confirming synergy.
>
> **Q4: Causal Language.** We accept this critique. Our Neural SEM with DAG+sparsity constraints learns a directed dependency structure within a Markov equivalence class [2], which is necessary but not sufficient for strict interventionist causality [3]. We will revise: "structurally coherent coalitions" replaces "causal coalitions," and "directed dependency graph" replaces "causal graph."
>
> **Q5: Hierarchical Extension.** NBCG extends naturally: apply Phase 1 at multiple thresholds $\tau_1 > \tau_2 > \cdots$ to obtain nested partitions, then perform multi-level Nash bargaining—coarse coalitions bargain globally, sub-coalitions locally—mirroring hierarchical cooperative games [4]. We will add this discussion.
>
> **References**
>
> [1] Von Luxburg, U. A tutorial on spectral clustering. *Statistics and Computing*, 17(4):395–416, 2007.
>
> [2] Zheng, X. et al. DAGs with NO TEARS. *NeurIPS*, 2018.
>
> [3] Pearl, J. *Causality*. Cambridge University Press, 2009.
>
> [4] Demange, G. Intermediate preferences and stable coalition structures. *J. Math. Econ.*, 23(1):45–58, 1994.

---

> > ### Author Rebuttal · Reviewer_F3CN · 2026-04-01
> >
> > After reading the rebuttal, I am more positive about the paper. The authors addressed several of my main concerns in a substantive way, especially by clarifying the causal framing, providing a more principled story for selecting the coalition structure, adding robustness evidence for graph perturbations, and giving a more convincing decomposition of the roles of learned coalitions and Nash optimization. These responses improve both my confidence in the method and my view of the paper’s overall clarity.
> >
> > I still think some limitations remain, particularly around the breadth of the new validation, the practical generality of the coalition-selection procedure, and the limited evidence on larger-scale settings. However, the rebuttal resolves enough of my original concerns that I am now slightly on the positive side overall.
> >
> > Accordingly, I will update my recommendation from 3 to 4.

---

### Decision · Program_Chairs · 2026-04-30

**Decision:**

Accept (regular)

**Comment:**

The paper studies long-tailed multi-label text classification. The Authors argue that the main challenge is not data scarcity per se, but dominant coalitions of frequent labels that may monopolize shared representations and gradient allocation, forcing rare labels to learn brittle shortcuts. The Authors propose NBCG, a two-phase framework that reformulates multi-label learning as a cooperative game among label coalitions. First, the label coalitions are created using Neural SEM. Then, the model is trained via a Nash bargaining objective that adaptively amplifies gradient emphasis on under-served coalitions. Experiments on four standard benchmarks show improvements in Rare-label F1.

The Reviewers acknowledged the importance of the problem and the potential of the proposed solution, in particular praising the creative synthesis of causal structure learning and cooperative game theory. Nevertheless, they also raised concerns regarding overstated causal language, insufficient rigor in the Pareto-efficiency claims, the absence of a principled criterion for selecting the number of coalitions N and the sparsification threshold $\tau$, absence of references to existing literature in which the cooperative game theory is applied to similar machine learning problems, and missing comparisons with recent multi-objective and bargaining-based baselines. The Authors' rebuttal substantially resolved these concerns and all Reviewers marked their concerns as fully resolved.

Beyond the Reviewer discussion, as AC I identified three additional issues. First, CCG (Fan et al., 2025) — the strongest baseline and the most closely related prior work, combining Neural SEM with game-theoretic optimization in a directly comparable fashion — is absent from the related work section; its omission is surprising and should be corrected. Second, the primary evaluation metric, Rare-label F1, is not formally defined: it is not stated whether F1 is micro- or macro-averaged over the selected tail labels, a distinction that materially affects interpretation. Third, the Nash bargaining objective is built on coalition-level BCE utilities, but no analysis connects this training surrogate to the Rare-label F1 metric used for evaluation; the rich literature on surrogate-based optimization of complex performance measures, including F-measures, is not discussed, and this gap should be addressed.

Given the strong reviewer consensus, the paper is above the acceptance bar; however, the Authors are required to address the three issues raised above in the camera-ready version, in particular by properly positioning CCG in the related work, formally defining the evaluation metric, and providing a discussion of the connection between the training objective and the reported performance measure.